# Space-Time Evolution Characteristics of Deformation and Failure of Surrounding Rock in Deep Soft Rock Roadway

**Xinfeng Wang [1,2,\*], Yiying Zhang [1], Qiao Zhang [1], Youyu Wei [1], Wengang Liu [1] and Tian Jiang [1]**

1   School of Environment and Resources, Xiangtan University, Xiangtan 411105, China
2   Emergency Management Research Center, Xiangtan University, Xiangtan 411105, China
\*   Correspondence: xfw2020@xtu.edu.cn

**Abstract:** In view of the problems of large deformation of surrounding rock, high in-situ stress, and serious soft fracture of rock stratum in deep soft rock roadway, the instability deformation failure mode of deep soft rock roadway is analyzed theoretically. The FLAC$^{3D}$ software is used to establish a three-dimensional numerical model of surrounding rock damage under load, and to study the displacement, stress, and plastic expansion process of damage and failure evolution in the surrounding rock of the roadway. The mechanical response mechanism of deep soft rock roadway surrounding rock bending deformation, elastoplastic transformation, and unloading failure is verified by MATLAB numerical analysis, and the space-time evolution characteristics of soft rock deformation and failure are revealed. The results show that the surrounding rock of deep soft rock roadway has many failure modes, such as obvious displacement and deformation, high stress concentration, and intensified plastic transformation in the surrounding rock. The vertical stress in the surrounding rock is concentrated at the direct top and bottom, and the horizontal stress is concentrated at the roadway side and bottom; plastic deformation and failure first appeared at the roadway side, and then extended to other parts. The research conclusion provides an important reference for surrounding rock control and roof management of high-stress soft rock roadway under deep excavation disturbance.

**Keywords:** deep roadway; weak surrounding rock; numerical model; deformation failure; surrounding rock control

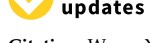



## 1. Introduction

With the needs of economic development, the society's demand for coal mining has also increased, and the shallow mineral resource reserves are far less abundant than the deep ones. So, it is imperative to develop deep mining vigorously. However, deep mining is different from the shallow one. Affected by the complex operating conditions of three highs and one disturbance (high geostress, high geotemperature, high osmotic pressure, and strong mining disturbance), the surrounding rock of the deep roadway is highly susceptible to deform under the action of high in-situ stress, high permeable water pressure, and groundwater softening. At the same time, in terms of supporting structure, the problems of supporting failure and the inability to exert the self-bearing capacity of the surrounding rock due to the improper selection of the supporting structure often occur.

For the research on the deformation and failure characteristics of the surrounding rock and the stability control technology of the deep soft rock roadway, many scholars at home and abroad have obtained a series of research results combining theoretical analysis and large amounts of engineering practices. Zuo Jianping and Wang Hui et al. [1–3] explored the destabilization and deformation characteristics, damage evolution law, and dynamic response mechanism of soft surrounding rocks in deep tunnels under the action of excavation and unloading disturbance by means of theoretical analysis and numerical simulation, revealing the spatial characteristics and scale effects of soft rock damage destruction. The

team of academician He Manchao [4–9] combined the classical theories of elastic-plastic yield criterion, theoretical mechanics, and engineering geology to conduct the research on the systematic characteristics of deformation and damage of deep soft rock roadway surrounding rock for the analysis of causes, classification of disaster evolution, and control technology, which pointed out the direction for the design of soft rock support. Against the current situation of large section chambers in deep mines with intense deformation and severe flaky gangs, Yuan Yong et al. [10] established a structural model to systematically reflect the stress-displacement multifactor characterization of the column mechanics of the roadway gang column, and analyzed the influence of mining parameters on the evolution of column displacement and stress. Huang Yaoguang [11] analyzed the distribution and evolution characteristics of the plastic damage zone of the surrounding rock of unequal-pressure circular roadway under deep high ground stress and obtained the time-dependent characteristics and optimized location of grouting reinforcement. Meng Qingbin et al. [12] studied the geomechanical characteristics of high-stress soft rock roadway by using the viscoelastic rheological model, and concluded that the deformation of the roadway surrounding rock presents the rheological characteristics of attenuated deformation and stable deformation. Ma Yuan and Zuo Yujun [13,14] explored the dynamic characteristics and spatio-temporal evolution law of the mobile deformation of deep soft rock under excavation disturbance conditions by using RFPA simulation software and field actual measurements. Yang Xiaojie et al. [15] investigated the mechanical mechanism of deformation and rupture of soft rock tunnel enclosure and proposed the coupled support technology of "bolting with wire mesh—grouting—concrete reinforcement". Wang Qisheng [16] studied the damage range, displacement extension trend, and stress distribution characteristics of the elastic, plastic, and fracture zones of the surrounding rock before and after excavation of the deep soft rock roadway, and proposed the combined support technology of short anchor rod hanging net and bottom anchor reinforcement. Zhao Xufeng [17] studied the mechanical properties and deformation spatial and temporal effects during the construction of deep soft rock tunnels through the comparison and verification of field measurements and numerical simulations, and proposed the strengthening system and lining measures for the bearing performance of the surrounding rock in deep buried soft rock tunnels.

Comprehensive analysis of the above literature reveals that the previous research on the deformation and damage characteristics and mechanical evolution law of deep soft rock roadways focuses more on the theoretical elaboration and macroscopic exploration of the mechanical behavior of deep soft rock damage, and lacks detailed demonstration and simulation inversion of the direct causative factors, internalization mechanisms, and damage modes affecting the deformation and instability of deep soft rock. There is still a need for further strengthening in generalizing and summarizing the mechanical evolution effects of dynamic damage destruction of deep soft rock under the coupling effect of multiple fields of the displacement field, stress field, and fracture field. How to use numerical analysis methods and numerical simulation software to deduce and show the dynamic response process of bending deformation, elastic-plastic transformation, and unloading damage of deep soft rock roadway surrounding rocks on the basis of realizing the mechanical model architecture and quantitative index analysis of deep soft rock unloading damage, and systematically represent the whole process of mechanical manifestation characteristics of deep soft surrounding rocks based on the multi-dimensional damage evolution in time, space, and scale from macroscopic and microscopic multiple levels, is a topic worthy of in-depth study. This paper focuses on the spatial and temporal evolution process of the deep soft rock roadway enclosure, integrating various research methods such as theoretical analysis, numerical simulation, and numerical operation, and establishing a numerical analysis model under the background conditions of a high-stress excavation project, which therefore summarizes and explores the possible evolution law of deep roadway model. The authors use Matlab software to draw the three-dimensional evolution cloud map of the bending moment and stress distribution of the roadway roof model, so as to reveal the deformation mechanism of the unloading disturbance of the

surrounding rock in a deep excavation roadway and to obtain the spatial and temporal evolution characteristics of deformation and failure of the surrounding rock in a deep soft rock roadway. The above exploration can provide a reference for the stability control of the surrounding rock in deep soft rock roadways.

## 2. Deformation and Failure Mode of Deep Soft Rock Roadway

### 2.1. Mechanical Model Criterion

There will be a succession of alterations as the roadway experiences the effects of mining and other factors. In order to observe the deformation characteristics of the surrounding rock during the roadway mining process, a simple mechanical model of the roadway is established, as shown in Figure 1. After the excavation of the roadway, the stress on the surrounding rock primarily comes from the gravity stress of the overlying strata, which can be viewed as a uniform load acting on the roof plate. According to the force characteristics of the surrounding rock, the roof and the gang can be simplified as a simply supported beam model, which can be regarded as an elastic-perfectly plastic material. Therefore, the roof surrounding rock is taken as the research object here. The surrounding rock load of the gang has a specific proportional relationship with the roof plate, which can be handled by analogy.

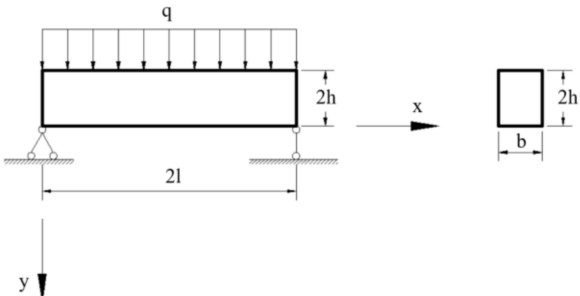

**Figure 1.** Simplified mechanical model of the roof beam.

Consider this model as a plane strain problem. The roof plate simply supported beam is loaded by a uniform load of size q, the length of the beam is 2l, the size of the cross-section is $b \times 2h$, and the coordinate system is established as Figure 1. The buried depth of the roadway is H, and the average density of the overlying rock layer is $\rho$. Thus, the gravity stress of the surrounding rock can be calculated as $p = \rho g H$. Other relevant mechanical parameters of the surrounding rock are shown in Table 1.

**Table 1.** Table of mechanical parameters of the surrounding rock.

| Elastic Modulus E/GPa | Poisson Ratio $\mu$ | Compressive Strength $\sigma_c$/MPa | Tensile Strength $\sigma_t$/MPa |
|---|---|---|---|
| 0.85 | 0.21 | 19 | 1.17 |

The roof plate transits from the elastic state to the plastic state when the bending moment of the roof plate beam is more than the bending moment at the elastic limit. The simply supported beam under uniform load attains the maximum bending moment and deflection at the mid-span position [18]. The plastic state bending moment equation of the roof beam in the plastic state is as follows:

$$M = \frac{3}{4}M_e\left(1 + \frac{1}{\beta}\right) \tag{1}$$

In Equation (1), $M$ is the bending moment of the top beam in the plastic state, $M_e$ is the bending moment of the top beam section in the elastic limit state, $\beta$ is the ratio of the ultimate tensile strength and ultimate compressive strength of the roof plate surrounding

rock, $\beta = 1/25$. Therefore, it can be obtained that the maximum bending moment in the mid-span of the roof plate surrounding rock is as follows:

$$M_p = \frac{39}{4}ql^2$$

The deflection equation for the plastic state of the roof beam [19] is as follows:

$$\begin{cases} w = \frac{q}{16bEh^3}\left(4lx^3 - x^4 - 8x\right) & 0 \le x < 0.08l, 1.92 \le x < 2l \\ w = \frac{5}{16}\frac{ql^4}{bEh^3}\left(\ln x - 1 - \ln l\right) & 0.08l \le x < 1.92l \end{cases} \tag{2}$$

In which, $w$ is the deflection of the top plate beam in the plastic state, $E$ is the elastic modulus, $x$ is the distance between any point on the beam and the left end point of the beam, and the relative relationship between $x$ and $l$ is shown in the coordinate system established in Figure 1.

The maximum mid-span deflection of the roof plate beam is as follows:

$$w_p = \frac{5}{16}\frac{ql^4}{bEh^3}$$

## 2.2. Significant Deformation of the Roadway Envelope

The convergence of the surrounding rock macroscopically characterizes the deformation of the surrounding rock caused by the roadway excavation to the inside, which is manifested by the downward movement of the roof plate, the convergence of gangs, and the upward movement of the floor. From Equation (2), it can be seen that the roof plate beam generates vertical downward displacement, i.e., the roof plate has downward settlement deformation, in which the deformation is the largest at the mid-line position, and the downward bending movement is the most obvious. Overall, the deformation is decreasing to both sides of the mid-line. The floor surrounding rock is similar to the macroscopic force, even if the calculation object of the Equations (1) and (2) is the roof plate beam. Therefore, the lateral force on gangs surrounding rock can be denoted by $\lambda p$ ($\lambda$ is the coefficient of lateral pressure, $p$ is the stress on the roof plate surrounding rock). Additionally, in the same way, both the floor and roadway gangs have the tendency to extrude and deform to the inside of the roadway.

The surrounding rock's instability and deformation first appear in the two gangs and bottom corners of the roadway. Excavation disturbance and high ground stress form loading and unloading effects on the surrounding rock in different directions. There are lateral unloading and vertical loading at the roadway gang, which superimpose with the vertical stress of the adjacent roadway. All of these cause the rock layer of the roadway gang to appear off-layer misalignment which macroscopically manifests as rib spalling. The deformation of the surrounding rock continues to occur with the transfer of internal stress of the rock mass, and shear stresses are transferred to the bottom corner, which leads those areas to be squeezed and broken. The rock deformation gradually extends from the upper part of the roadway gang to the top and shoulder. In the other direction, deformation extends from the lower part of the roadway gang and the bottom corner to the floor. The roof and floor generate inter-layer extrusion misalignment and converge to the middle of the roadway under the action of stress stretching. At the same time, due to the vertical contraction and deformation of the roadway gang, the constraint on the roof and floor is weakened, which intensifies the deformation of the roof and floor, and the macroscopic performance is the downward sinking of the roof and the upward dropsy of the floor.

## 2.3. Surrounding Rock Stress Concentration Appears

A rough analysis of the roof plate forces from a simplified mechanical model is made in terms of Equation (1), which shows that as the variable beam height h increases, the bending moment applied to the roof plate beam increases. The shear and bending moment

effects of the uniform load on the roof beam are shown in Figure 2. The beam has the largest bending moment at the center and is subjected to the most significant downward compression bending, whereas the shear force at the ends of the beam and the shoulder of the roof plate is the largest.

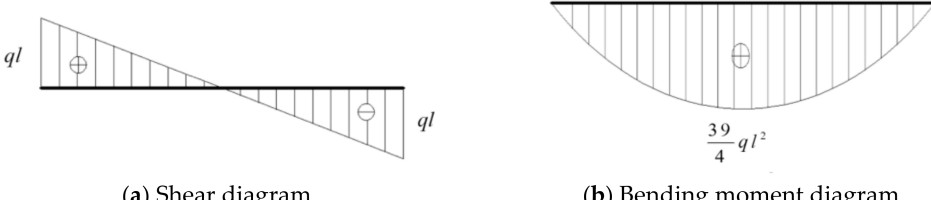

(**a**) Shear diagram  (**b**) Bending moment diagram

**Figure 2.** Schematic diagram of the distribution of shear force and bending moment in the roof beam.

From the perspective of mechanical environment analysis, this roadway model is buried at a depth of 800 m; the initial stress of virgin rock is high, and the self-weight stress has been higher than the ultimate compressive strength of the deep weak rock already. However, strong engineering disturbance stress act almost several times as much on the deep roadway as the stress in virgin rock mass [20–22], generating a strong additive and unloading effect on the rock mass, which makes the accumulation of energy in the rock mass suddenly released. In this case, the initial stress system in the surrounding rock of the deep roadway will change drastically.

From the perspective of force transfer evolution, a series of dynamic variations generate in the surrounding rock of deep soft rock roadway after the occurrence of mining disturbance. Once the roadway excavation begins, the stress balance of the virgin rock is immediately disrupted, the original tri-axial compression system is rapidly destabilized, and meanwhile, there is a sharp increase in the number of stress concentration points. Under the influence of one-way unloading, the stress of the surrounding rock is concentrated in the vertical direction and unloaded in the horizontal direction. For the rock layer in the shallow part of the roadway gang, the vertical stress is concentrated; roadway gangs and the shoulder and floor corner at the intersection of the roof-floor plate are more vulnerable, which is also the key area of stress concentration.

*2.4. Internal Elastic-Plastic Failure Transition of the Surrounding Rock*

The Mohr-Coulomb yield criterion [23] is expressed as:

$$f_s = \sigma_3 - \sigma_1 \frac{1 - \sin\varphi}{1 + \sin\varphi} + 2c\sqrt{\frac{1 - \sin\varphi}{1 + \sin\varphi}} \tag{3}$$

In the formula, $\sigma_3$ is the maximum principal stress, $\sigma_1$ is the minimum principal stress, $c$ is the cohesive force, and $\varphi$ is the angle of internal friction.

Assuming that there is a plastic zone distributed in the immediate top and above rock, the width of which is $X_p$. Then, combine the M-C yield criterion and the stress balance principle with a micro-element of plastic zone for analysis and calculation, we can derive the plastic zone width calculation formula [24]:

$$X_p = \frac{Xh}{A\tan\varphi_0} \tag{4}$$

In the Equation (4),

$$X = \ln\left[\frac{\frac{2(p\tan\varphi_0 + c_0)}{NA\tan\varphi_0 + c_0}X + \frac{pA\tan^2\varphi_0}{m(NA\tan\varphi_0 + c_0)} + 1}{X + 1}\right] \tag{5}$$

In the above expression, $c_0$ is the cohesive force of the rock formation, $\varphi_0$ is the angle of internal friction at the interface between the roof and coal seam, $\varphi$ is the internal friction angle of the rock body, $p$ is the vertical primary rock pressure, $N = \frac{2c_0 \cos \varphi_0}{1+\sin \varphi_0}$, $A = \frac{1+\sin \varphi}{1-\sin \varphi}$, $A$ is the ratio of the peak vertical stress to the horizontal stress in the ultimate equilibrium state, $h$ is the micro-element height of the plastic zone of the roof plate, l is the length of the roof plate, and m is the depth-span ratio of the roof beam, which is $m = h/l$ in this case.

From Equation (4), the state in which the plastic deformation of the surrounding rock in ultimate equilibrium is located is related to the spatial and temporal evolution of the vertical and horizontal stresses. It can be seen that the width of the plastic zone is related to rock friction angle, cohesion and depth-span ratio to some extent. The transformation of the elastic-plastic zone within the surrounding rock of the deep soft rock roadway is an evolutionary process that develops gradually with time. After excavating the deep roadway, the surrounding rock is rapidly destabilized and deformed, showing significant long-time rheology. The strong disturbing influence of the working face advance and the high ground stress under high buried depth work together to make the stress path within the rock body change violently. The surrounding rock structure appears to slip continuously on the whole, and a large amount of elastic strain energy is released inside the surrounding rock. The stress is transmitted along the fracture surface generated by the rock slip to the shallow rock, and the high coupling action stress creates a large number of tiny fractures in the shallow rock, resulting in the deterioration of the surrounding rock. The rock with a wide range of physical deterioration is transformed from the elastic zone to the plastic zone. In the evolutionary process, the fractures in the rock are continuously expanded with the transfer of stress, the rheology of the surrounding rock is intensified, the staggering deformation between the rock layers is produced, the fractures in the rock are developed, causing the rock to break and swell, and the expansion of the plastic zone is further enlarged.

## 3. Deformation and Failure Characteristics of Deep Soft Rock Roadways

### 3.1. Numerical Model Building

Based on a typical case of deep soft rock roadway project, a deep roadway model is established. In order to simplify the calculation and visually reflect the deformation and failure characteristics of the roadway, take the burial depth of the roadway as H = 800 m. The buried depth of the roadway is 800 m, the model size is $l \times d \times h = 80\text{m} \times 60\text{m} \times 37\text{m}$, the rectangular section size of the roadway is $l \times h = 5\text{m} \times 3.5\text{m}$, and the length of the roadway is 60 m. The vertical stress and horizontal stress are applied to simulate the high-stress environment. The model is a deep roadway under 800 m buried depth, so the self-weight stress of the surrounding rock should be considered. The average density of the overlying rock layer is $2.43 \times 10^3 \text{kg/m}^3$, the upper part gravity stress is $\rho g H$, the horizontal stress is $\lambda \rho g H$, and the coefficient of lateral pressure is ($\lambda = \mu/(1-\mu)$), in which μ is the rock Poisson's ratio. The simulated rock layer division and the mechanical parameters of the surrounding rock are shown in Table 2.

**Table 2.** Table of rock mechanical parameters.

| Rocks | Thickness/m | Density /(kg·m$^{-3}$) | Friction Angle/(°) | Modulus of Shear /GPa | Poisson's Ratio | Tensile Strength/MPa | Cohesive Force/MPa |
|---|---|---|---|---|---|---|---|
| siltstone | 10.00 | 2400 | 36 | 3.73 | 0.25 | 1.78 | 1.75 |
| Fine Sandstone | 6.00 | 2580 | 33 | 3.25 | 0.16 | 3.48 | 3.46 |
| Coal | 6.00 | 1350 | 27 | 0.35 | 0.21 | 1.17 | 1.19 |
| Sandy mudstone | 5.00 | 2370 | 31 | 1.43 | 0.26 | 1.81 | 1.96 |
| Mudstone | 10.00 | 1750 | 32 | 1.21 | 0.26 | 1.71 | 2.05 |

Referring to the typical deep soft rock roadway project, the support form set for the model here is the bolting and grouting support. Grouting behind the shaft is carried out at

the roof and the two gangs of the roadway, the distance between grouting holes is 2 × 1.5 m, and the thickness of grouting is 0.5 m; five and three high-strength bolts with a length of 2.5 m are installed at the roof and the gangs, respectively; the interval between bolts at the roof is 0.83 × 0.8 m, and the interval between bolts at the gangs is 0.88 × 0.8 m. The sketch of the roadway support system is shown in Figure 3, and the numerical model established is shown in Figure 4.

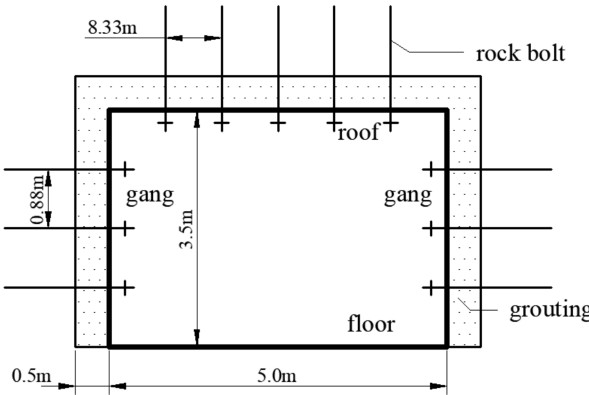

**Figure 3.** Roadway support sketch.

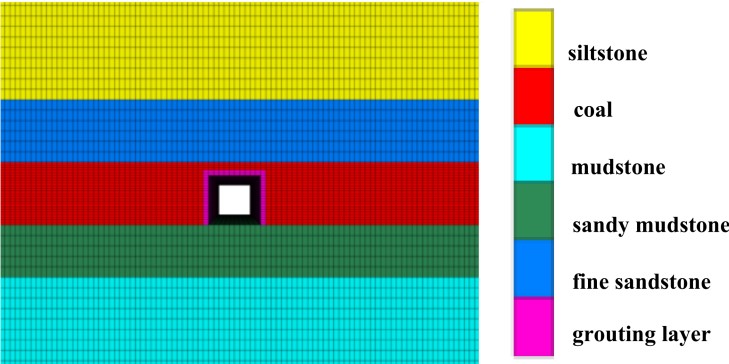

**Figure 4.** Numerical model.

Modeling with Flac³ᴰ software, setting parameters, and constraints to obtain the model under the original rock stress state, and solving again after excavation. A series of cloud diagrams that can characterize the deformation of the surrounding rock can be obtained, and the specific process can be represented by the flowchart in Figure 5.

The deformation and failure characteristics of the surrounding rock in the deep roadway greatly differ from those in the shallow roadway. The external reason is that the geomechanical environment in the deep roadway is more complex than that in the shallow roadway. In addition, due to the poor mechanical properties of deep weak rocks, the changes generated by the impact of engineering excavation disturbance will have more obvious deformation and failure characteristics. Next, the authors will analyze the features of the deformation and failure process of the deep roadway surrounding rock from three perspectives: deformation displacement, stress change and plastic zone distribution, combined with numerical simulation.

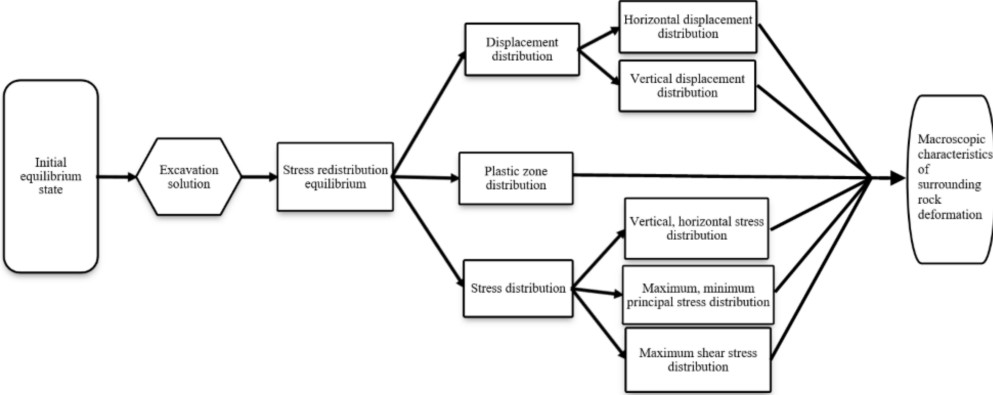

**Figure 5.** Numerical simulation flow chart.

*3.2. Deformation and Displacement Distribution Characteristics of the Surrounding Rock*

The deformation of the surrounding rock is a concentrated and significant manifestation of displacement changes in the roof and floor of the surrounding rock and the two gangs. The displacement changes in the surrounding rock are mainly manifested as the lift in the vertical direction and the slip in the horizontal direction. Therefore, the deformation displacement distribution characteristics of the surrounding rock can be explored by analyzing the displacement distribution clouds in vertical and horizontal directions, as shown in Figures 6 and 7.

First of all, analyzing Figure 6, we can see that the vertical displacement is symmetrically distributed relative to the center of the roadway. The displacement distribution at the roof plate is similar to the form of a flame, which gradually decreases from the center to both sides. The rate of displacement decrease gradually increases. Displacement gradually decreases from shallow to deep rock layers. The displacement value shown in the cloud diagram is negative, which means that the displacement at the roof plate points downward to the interior of the roadway. The vertical displacement distribution of the floor plate is like several concentric circles, and the displacement distribution and reduction characteristics of the shallow rock layer to the deep rock layer are similar to those of the roof plate; the displacement value at the floor is positive, which means that the displacement points upward to the interior of the roadway. Taking floor deformation as an example, the numerical model established in this paper is based on the engineering case without floor support, and the lack of floor support causes the stress and displacement changes to appear on the floor first. This is due to the restriction of the support body of the roadway gang and roof that cannot immediately adapt to the sudden change in stress, which has caused obstacles to the adjustment of rock stress and deformation extension of the gang and roof surrounding rock, so the floor is the first to appear at the original rock stress release, which leads to the appearance of the deformation of the roadway floor bulge. Regarding the overall analysis, the rate of change in vertical displacement of the roof-bottom plate decreases significantly from shallow to deep rock layers, and its deformation shows significant time-dependent and rheological properties. In addition, the displacement of vertical settlement of the roof plate is significantly larger than the maximum displacement value of the floor plate under the effect of the self-weight of rock, and the range of severe vertical deformation produced by the roof plate is significantly more extensive than that of the roadway floor.

Analysis of Figure 7 shows that the displacement distribution of the surrounding rock of the roadway in the horizontal direction is symmetrical, concerning the centerline of the roadway. The displacement distribution at the roadway gang is circular, the displacement values on both sides are opposite in sign, the displacement direction points to the interior of the roadway, and the maximum displacement values are approximately equal. On the whole, the displacement change in the horizontal direction is mainly manifested as the convergence of the left and right gangs to the interior of the roadway. The broken deformation area of the roadway gang is mainly distributed in the shallow rock layer,

and the gang's rock layer near the roadway's interior is the major area of horizontal displacement variation.

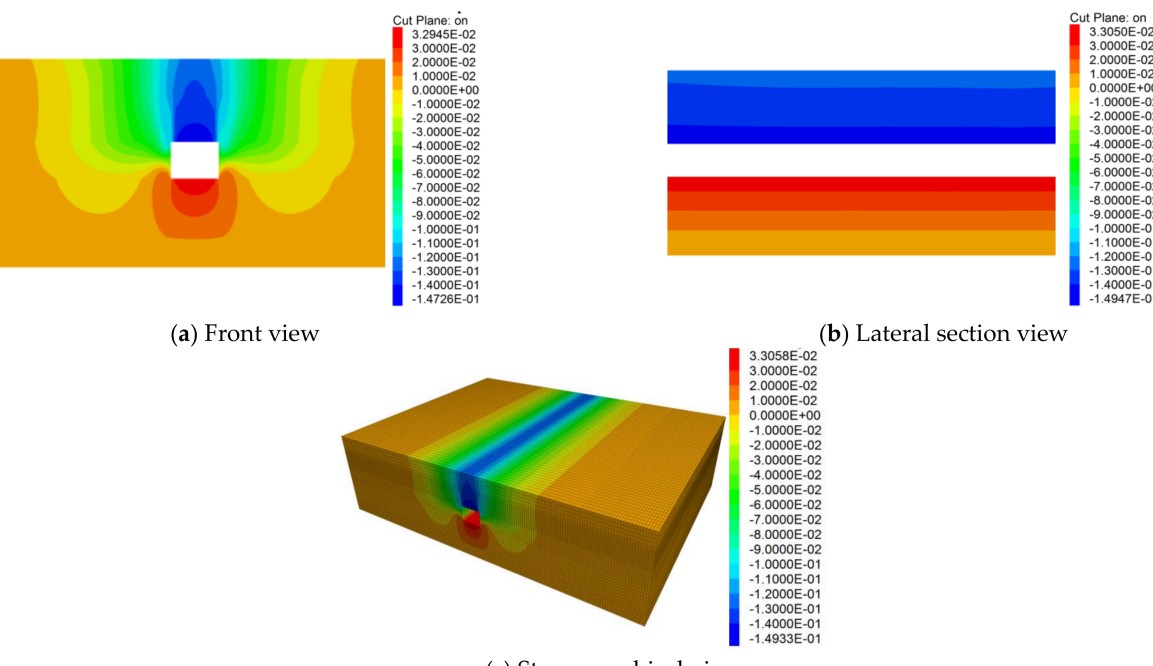

(**a**) Front view

(**b**) Lateral section view

(**c**) Stereographical view

**Figure 6.** Cloud maps of vertical displacement distribution.

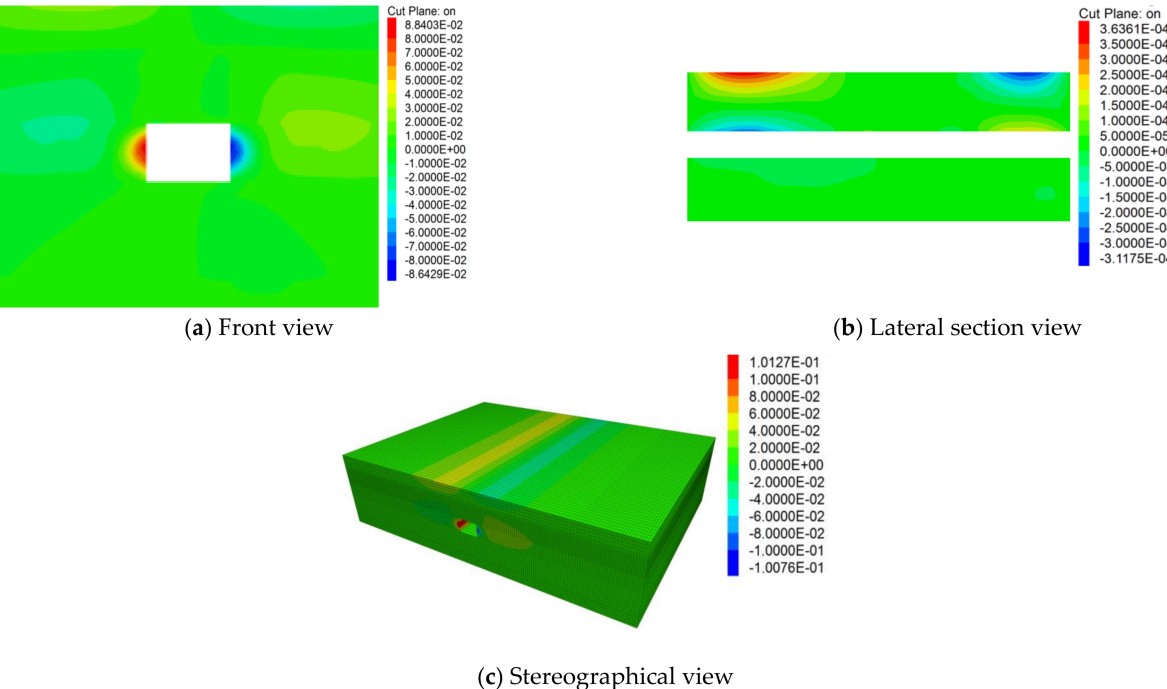

(**a**) Front view

(**b**) Lateral section view

(**c**) Stereographical view

**Figure 7.** Cloud maps of horizontal displacement distribution.

By recording the trend of the variation of the displacement of the surrounding rock with the advance of the working face, the deformation of the surrounding rock can be further characterized. Monitoring points were set at the roof, floor, and the right gang surrounding rock of the roadway model to inspect the variations of the surrounding rock displacement, and the displacement values of the monitoring points were exported to make a line graph so that we can observe the trend of its variation intuitively, as shown

in Figure 8. It can be found that as the working face advances, the displacement changes in the roof plate, floor plate, and roadway gang are generally on an upward trend. The most significant deformation in the monitoring process is the downward movement of the roof plate, with a maximum displacement value of 95.8 mm. Among the three main deformations, the convergence of the gang is the smallest, with a maximum displacement value of 49.1 mm. The deformation of the bottom plate rises between the roof plate and the two gangs, with a maximum deformation value of 65.4 mm in the process of monitoring. After that, the deformation of the roof and floor tends to stagnate, and the deformation of the top and bottom plates start to accelerate when they lose the vertical support of the roadway gangs, as shown in the 9880–11,080s time-step folding line in Figure 8. As the working face advances slow down, the change in the displacement of the gangs rapidly reaches stability, the rate of the top plate settling and bottom plate bulging is higher, and the phenomenon of roof sinking and floor bulging is significant. With the continuous advancement of the working face, the displacement of the surrounding rock still continues to increase, but the growth rate slows down significantly. Up to the late stage of monitoring, the deformation and displacement of the bottom plate tends to be stable, whereas the displacement of the top plate still has a slightly rising trend.

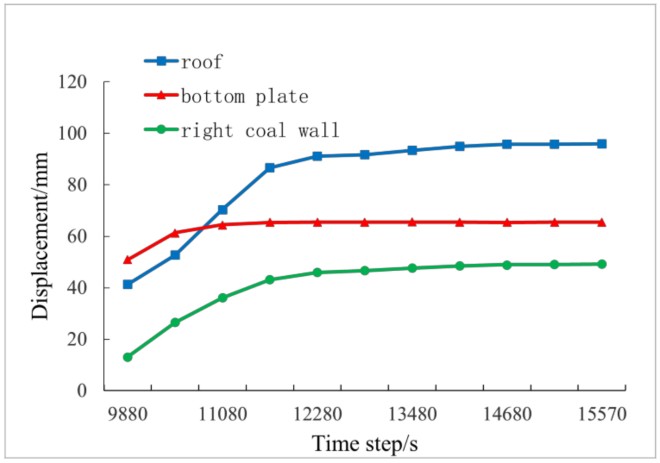

**Figure 8.** Displacement curve of surrounding rock deformation.

Based on the above analysis, the surrounding rock's external deformation characteristics are the roof plate's downward displacement, the shifting of the roadway gang, and the heave of the bottom slab. The distribution of deformation displacement in space presents the following characteristics: firstly, the vertical and horizontal displacement distribution of the surrounding rock is approximately symmetrical, showing significant currency and rheology properties; secondly, among the three main deformations, the top slab produces the most apparent displacement changes and the widest deformation range, followed by the bottom slab deformation; finally, under the influence of the factor of time, the range of deformation and expansion of the surrounding rock is increasing from the excavation of the roadway to the subsequent period of time, the surrounding rock will produce continuous deformation, and when a certain degree is reached, the amount of deformation of the surrounding rock tends to stabilize. It should be pointed out that if the crushed and expanded rock body further expands, then the surrounding rock is likely to produce collapse, flake gang, and other serious damage. The overall displacement of the surrounding rock continues to increase with the advance of the working face until it reaches a stable state.

### 3.3. Characteristics of Surrounding Rock Stress Distribution

During the excavation process, the stress state of the surrounding rock is always adjusted and continuously developed. Therefore, in order to investigate the evolution

characteristics of the stress during the deformation and failure process of the surrounding rock, the authors simulated and exported the stress distribution clouds of the surrounding rock in the vertical and horizontal directions, the maximum and minimum principal stress distribution clouds and the maximum shear stress distribution clouds, as shown in Figures 9–11.

Analyzing Figures 9 and 11b, the vertical stress field is approximately symmetrically distributed relative to the rectangular section, and the distribution pattern is relatively regular. The stress values at the top and bottom plate are positive, so the mechanical action is expressed as tensile stress, which spreads to the deep rock layer and shows a trend of gradually declining along the radial direction from shallow to deep until it is transformed into the compressive stress. On both sides of the roadway, the compressive stresses are concentrated in a wide range of surrounding rocks near gangs, extending to the lower part of the roadway and deeper layers in a cicada wing shape. The stresses gradually decrease during the extension process, and the rate of stress reduction gradually slows down.

Analyzing Figures 10 and 11a, the horizontal stress field distribution is approximately symmetrical; the horizontal stress on the top slab and the shallow rock above is mainly expressed as tensile stress. As for the evolution of tensile stress, it expands to the deep rock layer and gradually evolves into compressive stress, which will concentrate in the deep rock range of the top slab surrounding rock at the final of evolution. Combined with the analysis in Figure 11, it can be seen that at the intersection of the upper and lower parts, such as the shoulder, alternating tensile, and compressive stresses are acting on the surrounding rock, and the shear stresses are concentrated. Overall, the distribution of horizontal forces on the surrounding rock is not even, the stress value is relatively small compared to the vertical direction stress, and the degree of influence on the deformation of the surrounding rock is below the vertical stress.

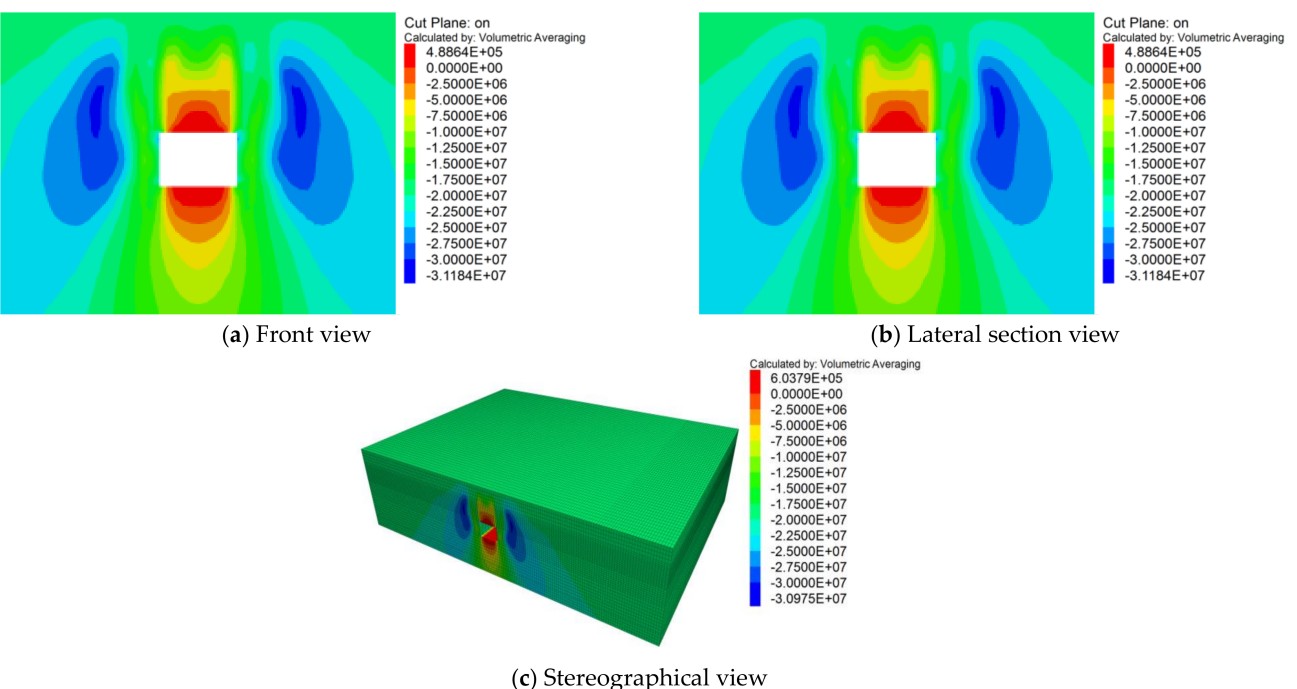

(**a**) Front view

(**b**) Lateral section view

(**c**) Stereographical view

**Figure 9.** Cloud maps of vertical stress distribution.

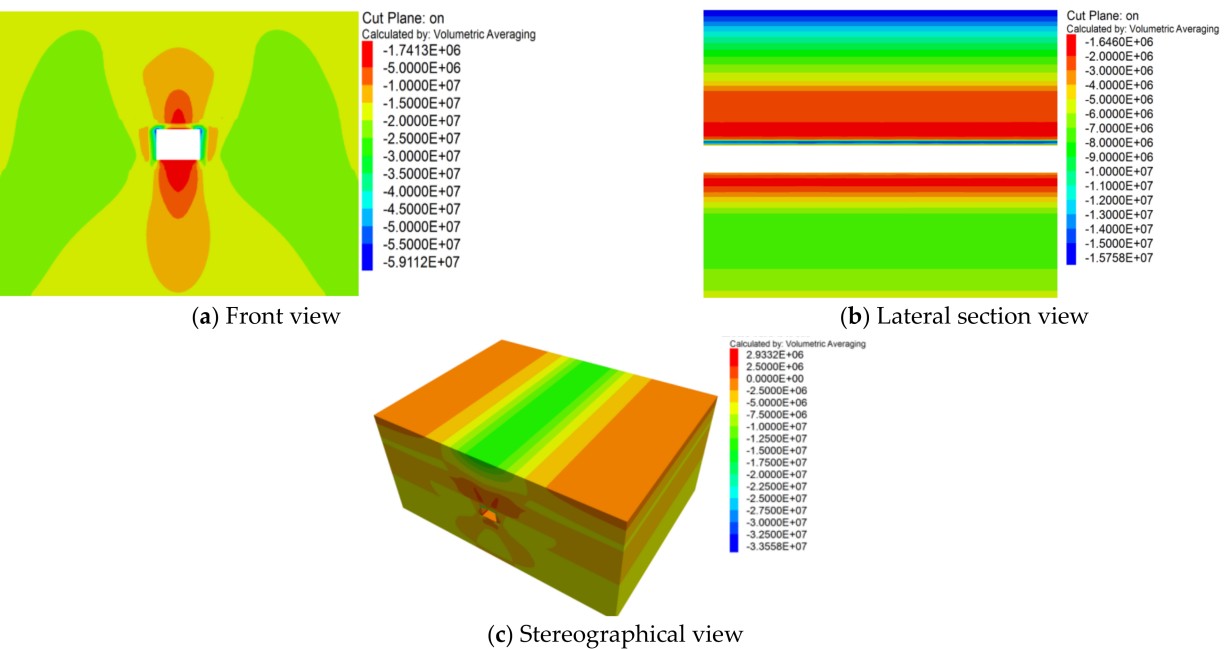

(**a**) Front view  (**b**) Lateral section view

(**c**) Stereographical view

**Figure 10.** Cloud maps of horizontal stress distribution.

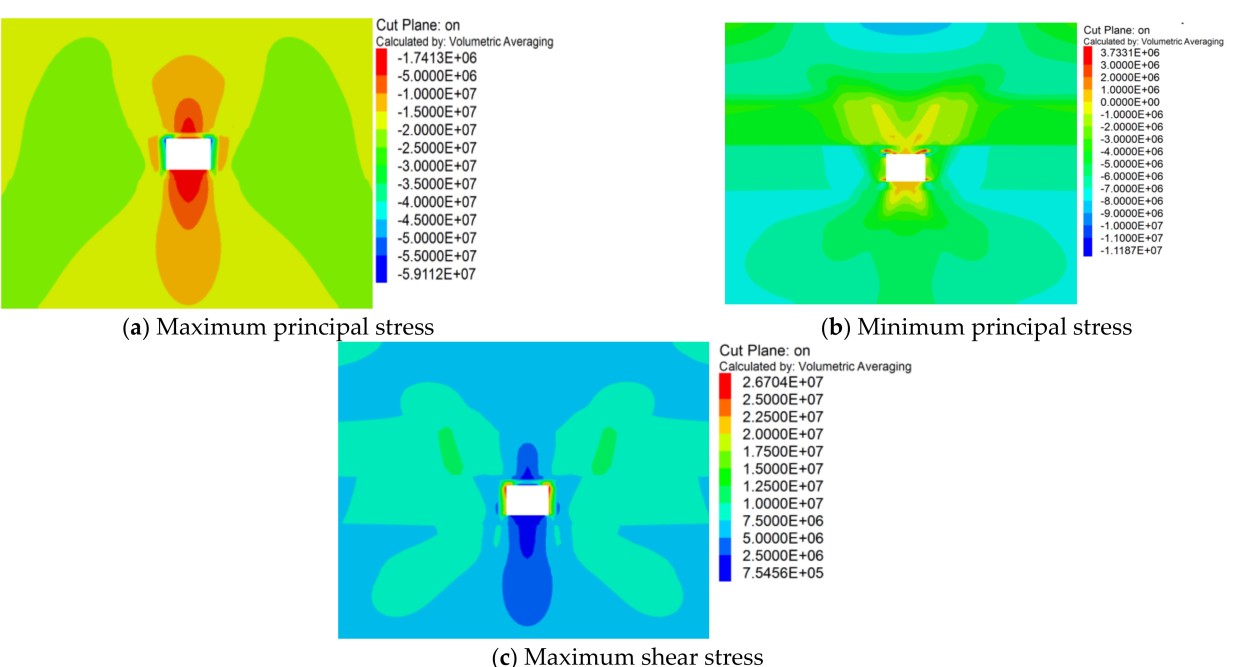

(**a**) Maximum principal stress  (**b**) Minimum principal stress

(**c**) Maximum shear stress

**Figure 11.** Other stress distribution cloud maps.

The comprehensive analysis of the stress distribution characteristics of the surrounding rock shows that the stress loading and unloading effect caused by roadway excavation is divided into two parts: (1) a part of the high stress in the rock body is released into the shallow surrounding rock in the form of energy, which causes deformation of the surrounding rock in a short period; (2) the release of another part of the high stress causes adjustment and redistribution of stress in the surrounding rock and causes stress accumulation in the local range. The excavation unloading makes the unilateral unloading of the surrounding rock into an unbalanced discontinuous stress body, and an extended evolution in space and time occurs. The process mainly shows the following characteristics: Firstly, the stress distribution in the surrounding rock is overall approximately symmetrical, with a regular distribution pattern. The vertical stress is mainly concentrated at the immediate roof and

immediate floor, and the stress value is high, so that it has a significantly destructive effect on the rocks around the roof and floor. Secondly, the horizontal stress concentration at the immediate floor is the largest, and the stress action range is also the widest, for which the damage by tension is more serious. Finally, the extension range of the vertical stress field is wider than the horizontal stress field, and the stress value is slightly larger than the horizontal stress. The stresses in the vertical and horizontal directions act together on the surrounding rock leading to compression-shear rupture as well as tensile deformation.

### 3.4. Failure Characteristics of the Plastic Zone of the Surrounding Rock

In the above-listed equation, $\sigma_3$ is the maximum principal stress, $\sigma_1$ is the minimum principal stress, c is the angle of internal friction, and $\varphi$ is the cohesion. Figure 12 shows the cloud map of the plastic zone distribution of the surrounding rock in the roadway model after excavation. The area divided by the green dashed line in Figure 12 is the fracture zone, and the stress curve of the rock in this range gradually tends to the Mohr's envelope and then reaches the yield point, at which the rock is compressed beyond its yield strength and forms a broken zone in this range. The rock adjacent to the broken zone is in a plastic state, whereas the stress curve of the rock in the deeper part does not reach the Mohr's envelope and is still in an elastic state.

The formation of the plastic zone is essentially the result of the change in internal stress in the surrounding rock and the attenuation of the structural strength of the rock mass. The dynamic adjustment of stress makes the surrounding rock an unstable mechanical environment, and the expansion and deformation of the rock body caused by the deformation of the surrounding rock and the misalignment of the structure surface fragmentation and expansion make the rock body strength decrease, and the combined effect of these two aspects leads to the formation of the plastic zone and the extension evolution to the deep part of the surrounding rock. As revealed in Figure 13, the broken and plastic rocks are mainly distributed in the bottom corner, the shoulder, the roadway gangs, and other surrounding parts. The plastic zone is similar to a symmetric butterfly-like distribution. After excavating the roadway, the lateral stress of the two gangs decreases sharply, and the vertical stress is concentrated, which makes the strength of the coal mass in the roadway gang decay and deteriorate. The coal mass in those areas enters into the post-peak damage state, generates a large volume of fissures in the rock mass, and transforms from the elastic zone to the plastic zone. The large area of coal mass of the middle roadway gang in the plastic zone occurs to break and slip, and then transforms into a broken state. Under the influence of shear stress, the coal body set to stagger and slip, and the broken zone gradually extends to the upper part of the roadway. It is not difficult to find that the deformation of the surrounding rock exhibits significant rheology. Over time, the loosely broken rock of the roadway gang continues to deform and expand to the interior of the roadway, which is macroscopically expressed as convergence towards the interior of the roadway. The shear stress is transferred to the bottom corner, shoulder fossa, and other local locations through the plastic zone, which causes plastic deformation.

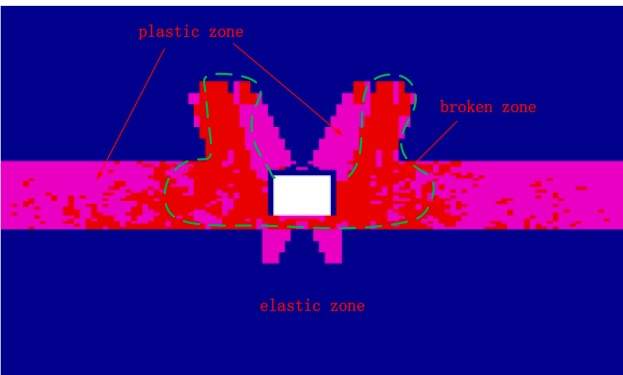

**Figure 12.** Cloud map of the plastic zone distribution of surrounding rock.

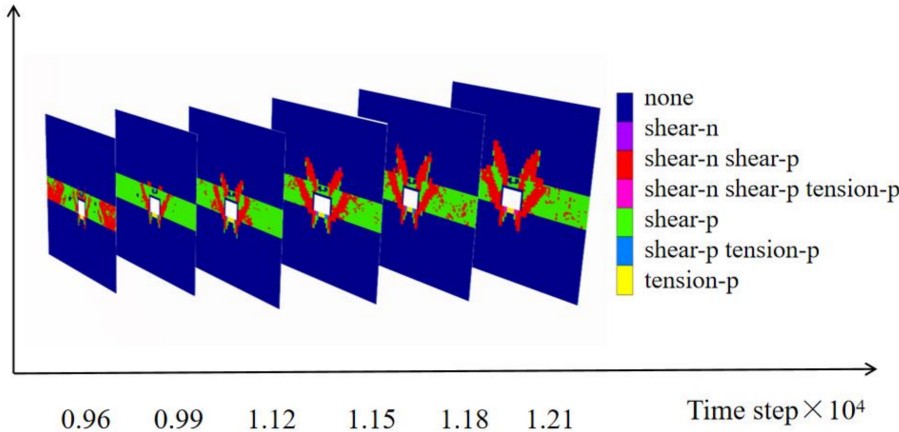

**Figure 13.** The development trend of plastic zone of surrounding rock.

By establishing a numerical model in the context of typical engineering cases, analyzing the deformation mechanism of the surrounding rock, summarizing the distribution characteristics of displacement, stress, and plastic zone, and obtaining the evolution trend and scale effect of deformation and damage of the surrounding rock in the deep soft rock roadway in two dimensions of time and space, as shown in Figures 14 to 15. At the early stage of roadway excavation, the surrounding rock stress field is extremely unstable, and the deformation of the surrounding rock is induced by the sudden change in stress in horizontal and vertical directions, and the surrounding rock is transformed from static elastic deformation to dynamic and drastic structural changes in rock layer, and the originally closed structural surface appears to slip. With the development of time, the deformation and damage rate of the surrounding rock decreases, the trend of increasing displacement weakens, and the plastic damage area still continues to extend to the deep part of the surrounding rock. In addition, before the surrounding rock stress field and deformation displacement field tends to stabilize, the deformation of the tunnel surrounding rock will continue for a long time, during which the development of the plastic zone of the surrounding rock gradually slows down, the development range eventually stabilizes, and the tunnel reaches a new state of stress equilibrium.

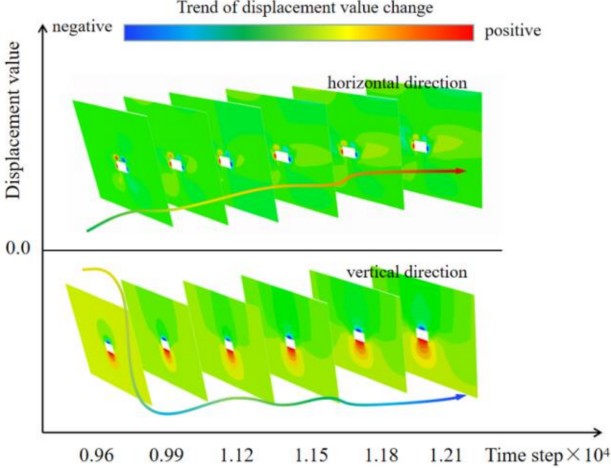

**Figure 14.** Space-time evolution trend of surrounding rock deformation.

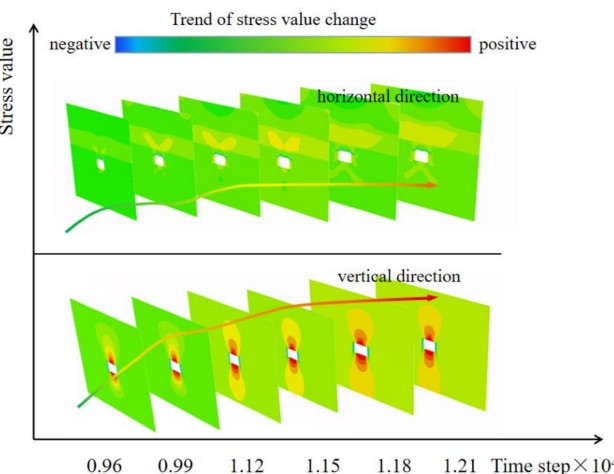

**Figure 15.** Space-time evolution trend of surrounding rock stress.

## 4. Engineering Simulation Validation

In order to further verify the analytical summary of deformation and failure characteristics of deep soft rock roadway surrounding rocks, the authors established a simplified mechanical model of the roof plate [25] surrounding rocks. The physical and mechanical background of the model is soft rock under a high buried depth. The bending deformation of this mechanical model is visualized by using Matlab finite element analysis software to analyze the deformation and failure characteristics of the roof plate surrounding rocks.

### 4.1. Engineering Model Building

The length of 5 m is taken along the tendency to the end of the roadway, and the length of the roadway section is 5 m; this roof unit is regarded as a thin rectangular plate subjected to uniform load in the upper part, $a \times b \times h = 5m \times 5m \times 0.9m$. The lower coal body of the top plate has been mined so that it cannot effectively support the roof plate. In addition, it is difficult to provide better restraint to the top plate since the coal bodies on both sides of the roadway are damaged by mining. Therefore, the restraint condition of the three sides of the thin rectangular plate is selected as simply supported. The coal body below the end of the roadway has a good restraint on the roof plate, which can be regarded as built in. Finally, the simplified mechanical model and coordinate system are established, as shown in Figure 16. The thin rectangular plate is set to be subjected to a uniform load q perpendicular to the upper plane, and the modulus of elasticity is E.

Combining the differential equations of the elastic surface $\nabla^2 \nabla^2 w = \frac{q}{D}$ with the Ritz method to solve the deflection expression $w(x, y)$ and then the bending moment and shear stress expressions.

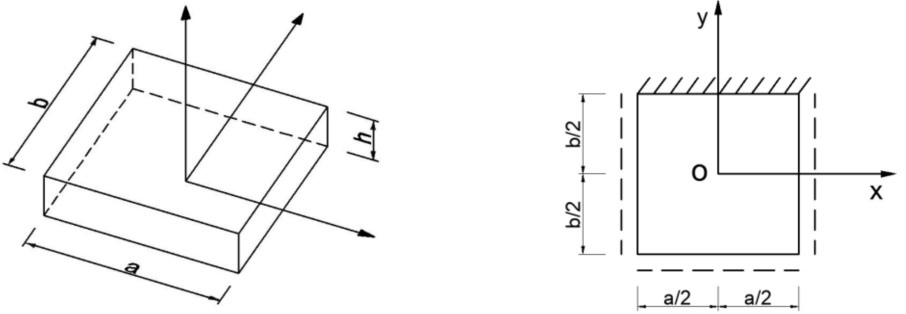

(**a**) Schematic diagram of a thin rectangular plate    (**b**) Schematic diagram of a rectangular plane

**Figure 16.** Simplified mechanical model of the roof plate.

*4.2. Analysis of Simulation Result*

1.  Roof plate deformation is obvious

Select the appropriate spacing distance, set the cell grid, and then transform the cell points on the rectangular plate into coordinate points in the *x-y* plane. Set the Poisson's ratio, stiffness, and other parameter values and then input the deflection expression. The last step is to execute the drawing command to obtain the deflection cloud map of this plate. Additionally, in order to improve the data comparability and compare the relative deformation of each point on the surface more intuitively, the deflection values of each point are projected on the *xOy* surface to obtain the cloud map projection, as shown in Figure 17. Figure 17 is the cloud map of the deflection distribution on the surface of the simplified mechanical model, in which the x-axis indicates the length along the strike; the y-axis indicates the length along the tendency; and the z-axis indicates the deflection value. Let the deflection downward be negative. Assuming that the thin plate only produces the deformation displacement in the vertical neutral surface, and does not produce the horizontal deflection, the cloud pattern is thus similar to the spatial pattern of the thin plate with a small deflection bending deformation.

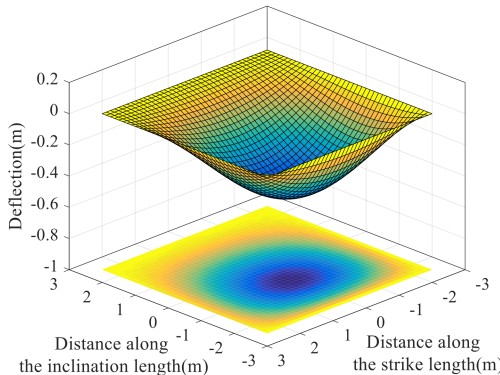

**Figure 17.** Deflection cloud map of thin plate.

Taking the deflection distribution characteristics at different locations as the entry point, Figure 17 shows that the deflection value is 0 at y = 2.5 m, and there is almost no bending deformation at the built-in support edge of the thin plate, which means that there is no deformation or tiny deformation of the roof plate at the end of the roadway. The deflection value in the middle of the simply supported edge gradually increases along the axis, and the deflection distribution cloud in the face is similar to multiple concentric rings, with the largest deflection value of 0.44 m in the face near the lowermost simply supported edge, and the minimum deflection value of approximately 0 in the four sides of the angle and the top built-in support edge. In general, the bending and sinking deformation of the roof plate surrounding rock is not uniform. The deformation of the corner situated at the cross position of the roof plate and the roadway gang is relatively lower, but the sinking in a certain range near the middle of the mined-out area is significant.

2.  High stress concentration inside the roof plate

The distribution cloud maps of $M_x$ and $M_y$ are obtained using the above method of drawing deflection clouds, as shown in Figure 18. The distribution and trend of the bending moment in Figure 18 show that $M_x$ gradually decreases along the inclination length from negative to positive. Then, it gradually increases after changing from positive to negative and achieves the maximum value at the center of the simply supported edge, which is opposite to the built-in supported edge. The trend of $M_y$ is similar to that of $M_x$, but the maximum value is obtained in the ellipse range adjacent to the opposite simply supported edge, and the overall numerical size of $M_y$ is higher than that of $M_x$.

In order to have a more integral understanding of the stress distribution characteristics, the following two equations are used to derive the expression for the shear force at the unit

point of the thin plate, which is expressed in a three-dimensional numerical Figure 18. The equations are as follows:

$$\begin{cases} V_x = -D\left(\frac{\partial^3 w}{\partial x^3} + (2-\mu)\frac{\partial^3 w}{\partial x \partial y^2}\right) \\ V_y = -D\left(\frac{\partial^3 w}{\partial y^3} + (2-\mu)\frac{\partial^3 w}{\partial x^2 \partial y}\right) \end{cases} \tag{6}$$

In which, $D$ is the bending stiffness of the sheet, $w$ is the deflection, and $\mu$ is the Poisson's ratio.

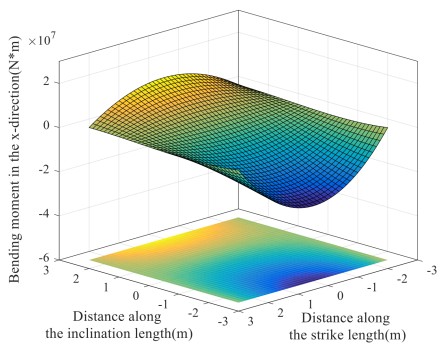

(**a**) Bending moment in the $x$-direction

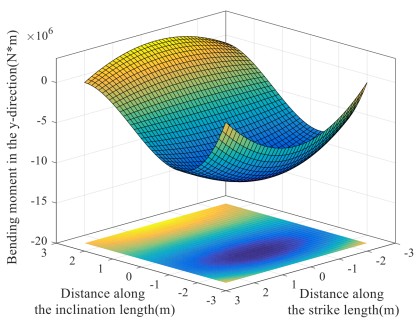

(**b**) Bending moment in the $y$-direction

**Figure 18.** Bending moment cloud maps of thin rectangular plate.

From Figure 19, we found that the shear value of $V_x$ is higher at the four corners, and the absolute values of shear at the two corners of the built-in supported side and simply supported side are slightly lower than the two simply supported side corners. The shear values at the same side corners are equal, and the signs are opposite. $V_y$ gains the maximum positive value at the center, and the absolute values of shear at the two corners of the lower side are the highest, with a negative direction.

Integrating the above analysis, the shear force is concentrated at the location of the shoulder where the roof plate meets the roadway gang. The stress concentration in the middle of the roof plate near the mining side is the highest, and the bending moment value is relatively higher, which performs the bending effect to the interior of the roadway for the roof plate.

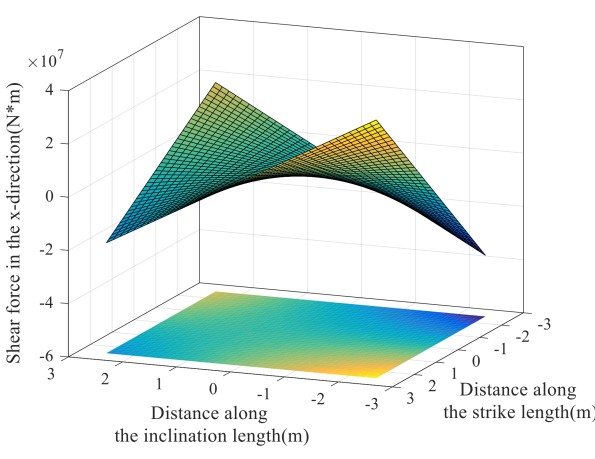

(**a**) Shear force in the $x$-direction

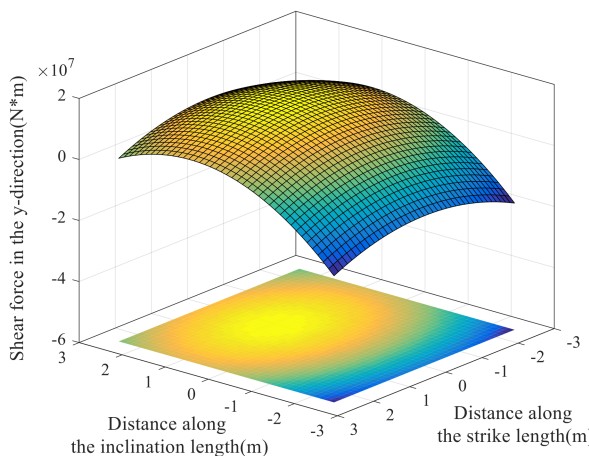

(**b**) Shear force in the $y$-direction

**Figure 19.** Shear force $V_x$, $V_y$ cloud maps of thin rectangular plate.

3. The roof plate rapidly transforms from elastic deformation to plastic deformation

The Mohr–Coulomb strength theory is being applied to the deformation and failure of this thin-plate model. Depending on the relative position of the Mohr stress circle and the shear strength envelope, the stress conditions of a particular stress body unit and whether its destruction has occurred can be judged. At the same time, this geometric relationship allows solving for the major and minor principal stresses in the limited equilibrium state of the stress body [26].

$$
\begin{cases}
\sigma_{1j} = \sigma_3 \tan^2\left(45° + \frac{\varphi}{2}\right) + 2c \tan\left(45° + \frac{\varphi}{2}\right) \\
\sigma_{3j} = \sigma_1 \tan^2\left(45° + \frac{\varphi}{2}\right) + 2c \tan\left(45° + \frac{\varphi}{2}\right)
\end{cases}
\tag{7}
$$

In the Equation (7), $\sigma_{1j}$ and $\sigma_{3j}$ denote the calculated value of the major and minor principal stresses in the limit equilibrium state of the stress body, c is the cohesive force, and $\varphi$ is the internal friction angle.

The expressions of maximum and minimum principal stress $\sigma_1$ and $\sigma_3$ are substituted into Equation (7) to obtain the calculated values of major and minor principal stresses. Then, we can obtain the difference between $\sigma_1$, $\sigma_3$, $\sigma_{1j}$ and $\sigma_{3j}$, using which to derive a set of surfaces of stress difference in three-dimensional space, and the iso-surfaces of stress difference are projected in the xoy plane. Thus, we can analyze the relative high and low relationship between maximum and minimum principal stresses and their calculated values in different parts of the thin plate, as well as analyze the distribution of plastic zone and the degree of plastic failure of the roof surrounding rock to a certain extent.

If $\sigma_{1j} < \sigma_1$ or $\sigma_{3j} < \sigma_3$, which means that the difference in the above Figure 19a is negative, and the difference in Figure 19b is positive, then shear failure has occurred at these locations. If $\sigma_{1j} = \sigma_1$ or $\sigma_{3j} = \sigma_3$, which means that the range marked as 0 in the above figure is in limiting equilibrium, then, conversely, the shear stress at this location has not reached the shear strength, and no failure has occurred. Observing Figure 17, it can be seen that the maximum and minimum principal stresses near the built-in supported side and at the angle of the lower two simply supported sides are equal to the critical values corresponding to the occurrence of shear failure. Observing Figure 20, it can be seen that the maximum and minimum principal stresses near the built-in supported side and at the angle of the lower two simply supported sides are equal to the critical values corresponding to the occurrence of shear failure. The mechanical properties are plastic, that is, corresponding to the top plate surrounding rock; the vulnerable plastic rock will generate fissures and cause deformation or even failure if there is a slight influence of engineering mining in this state. In addition, there are other areas where failure has occurred inside the model. For example, in the middle part of the thin rectangular plate near the side of the built-in supported, the shear stress reaches the shear strength, which means that the stress curve at each point on the roof plate surrounding rock at that location is tangent to the Mohr's envelope and failure has occurred. In the actual excavation process, the roof plate is subjected to high self-weight stress, when the excavation destroys the support constraint of the coal body below it, part of the roof plate develops from elastic deformation to plastic deformation, and the local range will sink obviously to the interior of the roadway. Such transformation is often rapid and transient, because the impact of mining is intense, and the internal stress of the surrounding rock increases rapidly over the surrounding rock's elastic limit, promoting the surrounding rock to transform into plastic deformation rapidly [27–29].

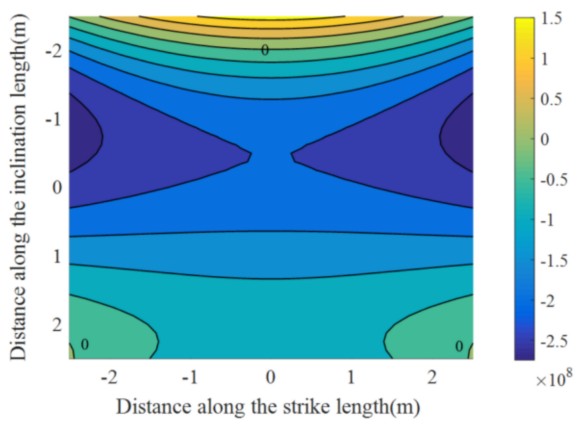

(**a**) Maximum principal stress difference distribution

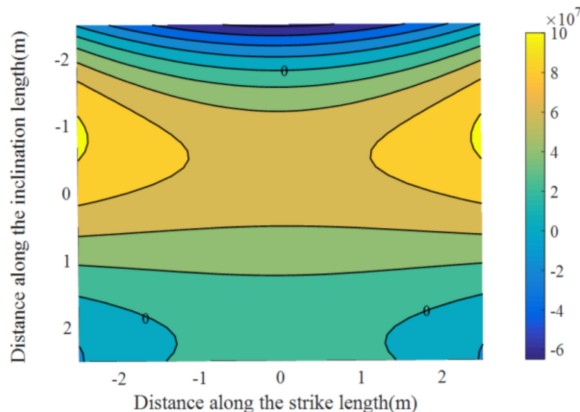

(**b**) Minimum principal stress difference distribution

**Figure 20.** Distribution of the difference between maximum and minimum principal stresses.

## 5. Conclusions

1. The simplified mechanical model analysis of the roof beam of the deep soft rock roadway shows that it has the characteristics of the maximum deflection in the middle of the roof beam, the concentration of shear stress at both ends of the roof beam, and the bending moment of the beam in the plastic state is obviously higher than the elastic limit bending moment. The deformation of the surrounding rock in the deep soft rock roadway has the features, such as significant displacement deformation, significant stress concentration, and sudden elastic-plastic failure, of the surrounding rock.

2. Through FLAC$^{3D}$ numerical simulation analysis, it is found that the deformation of the surrounding rock in the displacement, stress, and plastic zone distribution of the deep soft rock roadway presents the following characteristics: the surrounding rock produces an overall inward extrusion deformation; the vertical stress in the surrounding rock is concentrated at the immediate roof and the immediate bottom; the horizontal stress is concentrated at the roadway gangs and the floor; the plastic deformation damage occurs at the roadway gangs first, and then extends to other parts.

3. Using the simplified mechanical model of thin rectangular plate as an equivalent substitute for the deformation and failure of the roof plate surrounding rock under the setting background. The theoretical equations of deflection, bending moment, and stress of the model deformation are obtained by combining the elastic mechanics and geomechanics theories. The three-dimensional spatial demonstration cloud maps of the surrounding rock are derived and analyzed by Matlab software to reveal the spatial and temporal evolution law of the deformation and failure of the deep weak surrounding rock under the excavation and unloading perturbation. The simulation results above provide strong validation for the theoretical analysis and numerical simulation conclusions in the paper.

**Author Contributions:** Conceptualization, X.W.; data curation, Y.Z.; formal analysis, Q.Z. and Y.W.; investigation, W.L. and T.J. All authors have read and agreed to the published version of the manuscript.

**Funding:** This work is supported by the National Natural Science Foundation of China (Grant No. 51904266), Excellent youth project of Hunan Provincial Department of Education (Grant No. 21B0144), Hunan Youth Key Teacher Training Project and Research project on teaching reform of colleges and universities in Hunan Province in 2020 (Grant No. HNJG-2020-0231).

**Institutional Review Board Statement:** Not applicable.

**Informed Consent Statement:** Not applicable.

**Data Availability Statement:** The data used to support the findings of this study are included within the article.

**Conflicts of Interest:** The authors declare no conflict of interest.

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
