# Peer review of "Space-Time Evolution Characteristics of Deformation and Failure of Surrounding Rock in Deep Soft Rock Roadway"

_sustainability, doi:10.3390/su141912587_

Round 1

Author Response

Dear Editor,

Thank you for your letter on 5-September-2022, regarding the comments and recommendations of the paper entitled “Space-time evolution characteristics of deformation and failure of surrounding rock in deep soft rock roadway (ID 1866995). The comments are very valuable and have proved very helpful to us when revising and improving our manuscript. We have studied the comments carefully and have made the required revisions. Enclosed a highlighted copy of our revised manuscript in which the changes are marked in blue (see ‘sustainability-  1866995-Highlighted Copy’), and a detailed response letter to the editors and reviewers explaining the changes we have made (see ‘Response to reviewer1’).

Reviewer 2 Report

1、  The research background is not fully described. The research status of deformation and failure of surrounding rock in deep soft rock roadway is not well summarized, the literature is few and the focus is not clear. The literature review does not fit well with the research content of this paper and does not point out the shortcomings of the current research, so as to further summarize its own innovations, and further revisions are needed.

2、  In this paper, Flac and Matlab simulation methods are used for research, but the analysis of Flac numerical simulation is relatively simple. Compared with previous research by scholars, there is no in-depth and innovative elaboration on the temporal and spatial failure evolution characteristics of deep soft rock roadways. And is the determination of rock mass failure (shear stress) in Matlab analysis applicable to all types of rock mass?

3、  If there is engineering practice, it is recommended to add engineering site verification.

4、  The English expression needs to be further refined and revised. Here is one point that the expression in title 2.2 is incorrect.

5、  P150: If the roadway is buried at a depth of 800m, the self-weight stress is 20Mpa, and the compressive strength of the surrounding rock is 29Mpa, so the in-situ stress is not higher than the ultimate compressive strength of the rock mass, so the descriptions are contradictor.

6、  The roadway excavation position in the numerical model is in the middle of the coal seam, but the roadway excavation in the project site is mostly along the bottom or the top. Whether the numerical model conforms to the actual situation.

7、  P235: Format error

8、  P237: Figure 1” in title 3.2? Such obvious errors shall be checked in full text.

9、  Figures related to FLAC numerical simulation are not clear

Author Response

Dear Editor,

Thank you for your letter on 5-September-2022, regarding the comments and recommendations of the paper entitled “Space-time evolution characteristics of deformation and failure of surrounding rock in deep soft rock roadway (ID 1866995). The comments are very valuable and have proved very helpful to us when revising and improving our manuscript. We have studied the comments carefully and have made the required revisions. Enclosed a highlighted copy of our revised manuscript in which the changes are marked in blue (see ‘sustainability-  1866995-Highlighted Copy’), and a detailed response letter to the editors and reviewers explaining the changes we have made (see ‘Response to reviewer2’).

Reviewer 3 Report

The article addresses the problem of the stability of galleries located at great depth, under the conditions of exploitation of coal deposits in China. The analysis of the stability of these galleries is carried out both by analytical methods (beam theory) and numerical methods (FLAC 3D).

After studying this article we make the following observations:

-Case studies are very general and theoretical. Concrete, representative situations from coal mines in China should have been taken into analysis;

- The results of the analytical and numerical methods are treated individually, there being no comparative analysis of the results of the two types of methods;

- The results of the calculations are not validated by measurements performed in situ or in the laboratory;

-Drafting errors must be revised, such as: on lines 170-171, the definition of the parameters "C" and "fi" is reversed; on the scalar representations in all the figures resulting from the numerical modeling, the value of the parameters is not shown, etc.

 Although the topic presented in this article is interesting, taking into account the above observations, we believe that the research results should be reconsidered and the article should be completely rewritten in order to be publishable.

Author Response

Dear Editor,

Thank you for your letter on 5-September-2022, regarding the comments and recommendations of the paper entitled “Space-time evolution characteristics of deformation and failure of surrounding rock in deep soft rock roadway (ID 1866995). The comments are very valuable and have proved very helpful to us when revising and improving our manuscript. We have studied the comments carefully and have made the required revisions. Enclosed a highlighted copy of our revised manuscript in which the changes are marked in blue (see ‘sustainability- 1866995-Highlighted Copy’), and a detailed response letter to the editors and reviewers explaining the changes we have made (see ‘Response to reviewer3’).

Round 2

Author Response

Dear Editor,

Thank you for your letter on 15-September-2022, regarding the comments and recommendations of the paper entitled “Space-time evolution characteristics of deformation and failure of surrounding rock in deep soft rock roadway (ID 1866995). The comments are very valuable and have proved very helpful to us when revising and improving our manuscript. We have studied the comments carefully and have made the required revisions. Enclosed a highlighted copy of our revised manuscript (see ‘sustainability-1866995-revised paper’), and a detailed response letter to the editors and reviewers explaining the changes we have made (see ‘Response to reviewer’).

Reviewer 3 Report

The article has been significantly improved, which is why I propose to publish it in this form, following some editing and English language corrections.

Author Response

Thank you for your suggestions. The language in the article has been carefully revised and improved, and the content of the article has been enriched, which has enhanced the connotation and expression of the article.